# Clinical Factors Associated with COVID-19 Severity in Mexican Patients: Cross-Sectional Analysis from a Multicentric Hospital Study

**DOI:** 10.3390/healthcare9070895

**Published:** 2021-07-15

**Authors:** Joel Monárrez-Espino, Carolina Ivette Zubía-Nevárez, Lorena Reyes-Silva, Juan Pablo Castillo-Palencia, Julio Enrique Castañeda-Delgado, Ana Sofía Herrera van-Oostdam, Yamilé López-Hernández

**Affiliations:** 1Department of Health Research, Christus Muguerza del Parque Hospital, Chihuahua 31000, Mexico; lorenareyes@christus.mx; 2Vice Presidency of Health Sciences, Medical Specialties Program, University of Monterrey, San Pedro Garza García 66238, Mexico; carolina.zubia@udem.edu; 3General Hospital of Soledad de Graciano Sánchez, San Luis Potosí Health Services, Soledad de Graciano Sánchez 78435, Mexico; jpcastillopalencia@hotmail.com; 4Biomedical Research Unit, Mexican Institute of Social Security, Zacatecas 98000, Mexico; julioenrique_castaeda@yahoo.com.mx; 5Biochemistry Department, San Luis Potosí Autonomous University, San Luis Potosí 78290, Mexico; chromosomesxx.svo@gmail.com; 6Metabolomics and Proteomics Laboratory, Mexican Council of Science and Technology, Zacatecas Autonomous University, Zacatecas 98000, Mexico

**Keywords:** COVID-19, disease severity, Mexico, multivariate analysis, signs and symptoms

## Abstract

(1) Background: Latin America has been harshly hit by SARS-CoV-2, but reporting from this region is still incomplete. This study aimed at identifying and comparing clinical characteristics of patients with COVID-19 at different stages of disease severity. (2) Methods: Cross-sectional multicentric study. Individuals with nasopharyngeal PCR were categorized into four groups: (1) negative, (2) positive, not hospitalized, (3) positive, hospitalized with/without supplementary oxygen, and (4) positive, intubated. Clinical and laboratory data were compared, using group 1 as the reference. Multivariate multinomial logistic regression was used to compare adjusted odds ratios. (3) Results: Nine variables remained in the model, explaining 76% of the variability. Men had increased odds, from 1.90 (95%CI 0.87–4.15) in the comparison of 2 vs. 1, to 3.66 (1.12–11.9) in 4 vs. 1. Diabetes and obesity were strong predictors. For diabetes, the odds for groups 2, 3, and 4 were 1.56 (0.29–8.16), 12.8 (2.50–65.8), and 16.1 (2.87–90.2); for obesity, these were 0.79 (0.31–2.05), 3.38 (1.04–10.9), and 4.10 (1.16–14.4), respectively. Fever, myalgia/arthralgia, cough, dyspnea, and neutrophilia were associated with the more severe COVID-19 group. Anosmia/dysgeusia were more likely to occur in group 2 (25.5; 2.51–259). (4) Conclusion: The results point to relevant differences in clinical and laboratory features of COVID-19 by level of severity that can be used in medical practice.

## 1. Introduction

COVID-19 is an emergent viral infection responsible for the worst pandemic the world has seen since the Spanish Flu of 1918 [1]. One and half years after the first cases were reported, the illness has caused more than 3.8 million deaths worldwide, from which nearly 230 thousand have occurred in Mexico, the fourth most affected country in terms of absolute mortality after the United States, Brazil, and India [2].

The causative agent, SARS-CoV-2, is an RNA virus that enters the host cells by binding its spike protein to the angiotensin-converting enzyme 2 receptors, expressed in several body tissues (i.e., lung, liver, small intestine, esophagus, colon, and organs involved in blood pressure regulation). The virus then uses the host’s translation machinery to synthesize new virions that are released from the host cells to repeat the cycle [3]. During this process, the virus triggers an inflammatory response, accompanied by cytokine storms [4] and coagulopathies [5] if the infection is severe.

Respiratory distress occurs when the alveolar epithelial type 2 cells are involved, causing pneumonia [6], resulting in a reduced oxygen diffusion capacity [7] that often requires oxygen therapy [8]. If hypoxemia is acute and goes untreated, patients die, as it occurs in many low-income settings [9]. Critically ill patients at intensive care units (ICUs) usually die from multi-organ dysfunction with cardiac, renal, hepatic, hematologic, and neurologic involvement [6]. A large Chinese cohort of 72,314 showed that 81%, 14%, and 5% of COVID-19 patients receiving medical care presented a mild/moderate, severe, and critical illness, respectively [10].

While SARS-CoV-2 is transmitted like most other respiratory viruses, its impact on human health has been considerably more harmful, particularly when individuals are men, of advanced age, and with comorbidities such as diabetes (DM), hypertension (HTN), and obesity [11].

Since half of the symptomatic patients become infected from asymptomatic carriers [12], hygienic and social distancing measures have become the cornerstone of preventive efforts [13]. It has been estimated that 70% of infection transmission occurs during the incubation period (median = 7.2, 95th percentile = 15.1 days) [14]. There is also evidence indicating that a timely diagnosis and treatment can delay illness progression and improve prognosis [15].

Numerous symptoms are linked to COVID-19. General symptomatology includes headache, fever, myalgia, arthralgia, and fatigue; cough, sore throat, dyspnea, and chest pain are also frequent respiratory symptoms. Other less common symptoms include nausea, anosmia, dysgeusia, rhinorrhea, diarrhea, hyporexia, dizziness, confusion, skin rash, and hemoptysis [16,17,18].

Hematological and immunological markers have been used for the diagnosis and prognosis of COVID-19. Laboratory findings reflect the body’s immune response and are crucial for the understanding of the clinical course of the disease (e.g., viremia, defective host response, bacterial superinfection, cytokine storm, consumptive coagulopathy, impaired liver function, and organ damage). High C-reactive protein (CRP), sedimentation rate, lactate dehydrogenase, and serum ferritin (SF) have been frequently reported in COVID-19 patients. Elevated levels of D-dimer, high neutrophil counts, and high levels of aspartate transaminase (AST) and alanine transaminase (ALT) have also been reported in critically ill patients. On the other hand, low lymphocyte and platelet counts have been observed, indicating a defective host response and consumptive coagulopathy, respectively. In severe cases, studies have also documented increased procalcitonin (PCT), tumor necrosis factor alpha (TNF-α), interferon-γ-induced protein 10 (IP-10), monocyte chemoattractant protein 1 (MCP-1), chemokine (C-C motif), ligand 3 (CCL-3), and various interleukins (IL-2, IL-6, IL-7, IL-10) [19,20,21].

Computerized tomography (CT) has also become a major diagnostic and assessment tool, and it is now considered as a practical and reliable method in COVID-19 patients [22,23]. Typical radiologic findings include chest CT anomalies [24,25] (e.g., multifocal groundglass opacities with or without consolidation in lung regions close to visceral pleural surfaces, crazy paving, patterns compatible with organizing pneumonia, and thickened vessels with parenchymal abnormalities). Patients are assessed using the 1-5 scale COVID-19 Reporting and Data System (CO-RADS) to determine the likelihood of infection, with CO-RADS 4 and 5 considered to be diagnostic [26].

While Latin America has been a harshly hit region, with Brazil and Mexico topping the list in terms of mortality, the reporting of clinical data from these countries is still incomplete. Therefore, this study aimed at identifying and comparing the clinical characteristics of individuals attending three hospitals located in central and northern Mexico for COVID-19 diagnosis and/or treatment at different stages of disease severity.

## 2. Materials and Methods

### 2.1. Study Design

This was a cross-sectional multicentric study. Individuals with or without COVID-19 symptomatology attending the three participant hospitals for COVID-19 diagnosis or treatment were included. SARS-CoV-2 identification was performed using Reverse Transcription Polymerase Chain Reaction (PT-PCR) from a nasopharyngeal specimen using standard methods [27].

Table 1 summarizes the city and hospital characteristics, number of participants per group, period of data collection, inclusion criteria, and data obtained.

PCR-tested individuals were categorized into four mutually exclusive groups as follows: (1) PCR-negative, used as controls, (2) PCR-positive, not hospitalized, (3) PCR-positive, hospitalized with or without supplementary oxygen, and (4) PCR-positive, intubated at the intensive care unit (ICU).

In the capital city of Chihuahua (CHI), the sample included individuals aged >18 years who provided informed consent. Christus Muguerza Hospital is a private facility with 64 hospital and 15 ICU beds, where nearly 9000 patients are hospitalized every year; 19 hospital and 5 ICU beds were allocated to treat COVID-19 patients. Data were collected between 15 August and 1 December 2020, including blood samples for laboratory analyses. A total of 127 individuals completed the clinical questionnaire (groups 1 = 49, 2 = 29, 3 = 37, and 4 = 12). In addition, pulmonary computerized tomography (CT) scans were obtained for all patients hospitalized (groups 3 and 4).

The General Hospital from the municipality of Soledad, located in the Metropolitan area of San Luis Potosi (SLP), the state’s capital, is a public facility run by the Ministry of Health. Since it was a designated COVID-19 hospital, all 90 hospital and 8 ICU beds were used to treat infected patients. Clinical data were collected from September 15 to December 1 2020, for 55 individuals aged >18 years who provided informed consent. Blood samples for laboratory analyses were obtained from 43 patients (groups 1 = 5, 2 = 3, 3 = 39 and 4 = 8). No chest CT scans were available.

In the capital city of Zacatecas (ZAC), data were obtained from the General Hospital No. 1, a public facility run by the Mexican Institute of Social Security with 207 hospital beds (95 COVID, 5 ICU). Clinical and laboratory data from a subsample of 160 individuals aged 35–70 years (40 per group) were retrieved from the hospital’s electronic COVID-19 dataset of 1058 symptomatic patients tested for COVID-19 between 15 March and 1 November 2020. No radiologic information was available. Permission to extract the data from the files was obtained from hospital authorities.

Figure 1 shows the geographic location of the participant hospitals within Mexico, and compares total population, life expectancy, and COVID mortality with two countries from the region.

### 2.2. Data Collection

A questionnaire was used to collect sociodemographic data (i.e., age, sex, civil status, occupation, place of residence, and formal education), non-pathological medical data (i.e., smoking habits, alcohol intake, drug use, and physical activity), and pathological medical data, including treatment (i.e., DM2, HTN, overweight/obesity, chronic obstructive pulmonary disease—COPD, asthma, cancer, immunosuppression, chronic renal disease—CKD, allergies, and other infections/illnesses).

Dichotomous recounts, including the onset and treatment received, were recorded for the following signs and symptoms: fever, headache, myalgia, arthralgia, fatigue, irritability, chills, nausea, vomiting, rhinorrhea, cough, sore throat, dyspnea, chest pain, anosmia, dysgeusia, diarrhea, abdominal pain, and conjunctivitis.

Epidemiological data included questions about previous contact with a flu or known COVID-19 patient, flu vaccination, and travel within the last 15 days.

Lab data included conventional blood tests, but some immunological markers (e.g., PCT, CRP, SF, and IL-6) were also measured in hospitalized patients from CHI. Venous blood samples were collected at admission from the right arm using a hypodermic needle for a basic chemistry panel (i.e., glucose, creatinine, uric acid, cholesterol, and triglycerides) and blood count (i.e., hemoglobin, neutrophils, lymphocytes, monocytes, eosinophils, and platelets). For some patients from CHI, IgM and IgG results were also available.

Trained medical assistants were used to collect data from individuals from groups 1 and 2. For patients in groups 3 and 4, the data were obtained by trained physicians and medical residents.

### 2.3. Statistical Analyses

Frequencies and proportions were used to describe socio-demographic characteristics, main comorbidities, symptomatology, and epidemiological data for the study participants, stratified by hospital location and COVID-19 severity group. The proportions of individuals falling below or above the established reference standards for each laboratory measure were also computed; for some measurements (i.e., PCT, CRP, SF, IL-6, IgM, IgG, and CORADS), only data from CHI were available.

Crude odds ratios (ORs) with 95% confidence intervals (CIs) were computed using multinomial logistic regression. This is a method that generalizes logistic regression to predict the probabilities of the different possible outcomes of a categorically distributed dependent variable given a set of independent variables [28]. Here, COVID-19 grouping was used as the dependent variable, and the clinical and laboratory data available for the three hospitals were used as independent variables individually. The COVID-19 group 1 (i.e., PCR-negative controls) was used as a reference category.

A multivariate multinomial logistic model was also computed to produce adjusted ORs. The full model included all variables with a *p*-value of 0.10 or less in crude analyses. Backward elimination was used for variable selection. Only independent variables with at least one statistically significant (*p* < 0.05) category in the comparisons (i.e., groups 2, 3, and 4 vs. 1) remained in the final model. Therefore, age group, hospital location, HTN, fatigue, sore throat, chest pain, diarrhea, headache, lymphopenia, hypercreatininemia, thrombocytopenia, hyperglycemia, and anemia were excluded.

The Nagelkerke pseudo R^2^ statistic, ranging from 0 to 1, was used to provide an indication of the amount of variation in the dependent variable, explained by the regression model.

## 3. Results

### 3.1. General Characteristics of the Population by Hospital

The socio-demographic characteristics, main comorbidities, symptomatology, and epidemiological data of study participants, stratified by hospital location, are shown in Table 2.

Most individuals were male, irrespective of location, ranging from 55% in ZAC to 60.9% in SLP. The mean age in ZAC and SLP was identical (53 y), but 10 years lower in CHI, where nearly half of the participants were aged 20–40 years. Smoking prevalence varied markedly (SLP 3.6%, ZAC 9.4%, CHI 17.3%). The frequency of comorbidities also varied, especially for DM2 (CHI 9.4%, ZAC 23.8%, SLP 41.8%), but less so for HTN (CHI 22%, ZAC 33.1%, SLP 36.4%). Other illnesses were less prevalent.

For some variables, data were available only for the private hospital of CHI, where most patients had a spouse/partner (60.8%), were employed (77.8%), and reported a sedentary life (58.5%); in this group, the mean BMI was 29 kg/m^2^, with 38.3% of the participants having overweight and 40.2% obesity.

General symptoms such as fever, headache, myalgia, arthralgia, and fatigue were highly prevalent, affecting up to two-thirds of patients (e.g., headache in SLP and ZAC); it was noted that the prevalence of these symptoms was consistently lower in CHI. Respiratory symptoms such as cough and dyspnea were the most frequently seen, reported by 74.5% and 83.6% of the participants in SLP, respectively; sore throat and chest pain had the lowest prevalence. Patients from SLP reported respiratory symptoms more often than those from ZAC and CHI. Anosmia/dysgeusia and diarrhea had far lower frequencies in all hospitals (<15%).

The use of medications was only recorded in CHI, where 34.9% and 19% had received dexamethasone and baricitinib, respectively. Recent contact with a COVID case and travel within the last 15 days was mentioned by 57% and 40.6%, and by 11.9% and 16.8% of participants from CHI and ZAC, respectively. Influenza vaccination was also reported by 15.4%, 22.2% and 23.6% of the individuals from ZAC, CHI, and SLP, respectively.

### 3.2. Population Characteristics by Severity Groups

The socio-demographic characteristics, main comorbidities, and symptomatology for the study participants, stratified by severity groups, can be seen in Table 3.

Men were more affected as the COVID-19 severity group increased, ranging from 56.9% in group 2 to 68.3% in group 4. Patients’ age was also positively associated with COVID-19 severity group, with higher means in groups 3 and 4 (≈55 y); younger people tended to be classified in groups 1 and 2, and older people in groups 3 and 4.

DM2 and HTN were highly prevalent in group 3 with 37.1% and 35.3%, and group 4 with 33.3% and 50%, respectively. COPD/asthma and CKD were also more common in groups 3 and 4, but with a lower occurrence (520137%). The prevalence of obesity was 38.8% and 41.7% in groups 3 and 4, respectively.

Symptoms also varied across COVID-19 groups with consistently higher prevalence with higher COVID-19 severity for both general and respiratory symptoms.

### 3.3. Laboratory and Radiologic Analyses by Severity Group

Laboratory and radiologic findings for participants stratified by COVID-19 severity group are shown in Table 4.

Anemia prevalence was highest in groups 3 and 4 with ≈30%. The proportion of patients with thrombocytopenia and lymphopenia increased with COVID-19 group, being highest in groups 3 and 4. The prevalence of neutrophilia, hypercreatininemia, hyperglycemia, and hyperuricemia also increased with COVID-19 severity. PCT, CRP, SF, IL-6, as well as PCR, IgG/IgM, and CT scan were only available for some patients from CHI. Most patients from groups 3-4, in which SF and IL-6 were measured, had high values. The proportion of individuals tested with IgG-IgM positivity was low, ranging from 0 (IgG group 2) to 27% (IgG group 3).

Chest CT scans were taken in 49 patients of COVID-19 groups 3 (*n* = 37) and 4 (*n* = 12), who showed higher proportions of CO-RADS 5 and 6.

### 3.4. Crude Analysis

Table 5 shows crude ORs with 95% CI from multinomial logistic regression, with the severity groups as the dependent variable (group 1 as reference), and various selected factors as independent variables.

Most independent variables tested showed significantly increased ORs for either group, using group 1 as reference. Compared with women, being male was associated with a higher probability of falling into group 3 (2.22; 1.27–3.90) or 4 (2.25; 1.14–4.43). Similarly, ORs were high in groups 3 and 4 for those aged 46–65 years, and highest for those aged >65 years compared with controls aged 20–45 years. The odds for DM2, HTN, and obesity were also increased for groups 3 and 4. Neutropenia, lymphopenia, hyperglycemia, and hypercreatininemia showed elevated ORs in groups 3 and 4 too. Most symptoms were also associated with higher ORs as the COVID-19 severity group increased.

### 3.5. Adjusted Analysis

Nine variables remained in the final adjusted model, explaining 76% of the variability. All other variables had at least one statistically significant category (*p* < 0.05) across the comparison groups. Compared with women, adjusted ORs for men increased with COVID-19 severity from 1.90 in the comparison of 2 vs. 1 to 3.66 in the comparison of 4 vs. 1. DM2 and obesity were strong comorbidity predictors of COVID-19 severity group; for DM2, adjusted ORs for groups 2, 3, and 4 were 1.56 (0.29–8.16), 12.8 (2.50–65.8), and 16.1 (2.87–90.2), and for obesity, 0.79 (0.31–2.05), 3.38 (1.04–10.9), and 4.10 (1.16–14.4), respectively. Fever, myalgia/arthralgia, cough, dyspnea, and neutrophilia were more predictive as the COVID-19 severity increased. Conversely, patients with anosmia/dysgeusia followed a downtrend pattern with higher odds in groups 2 (25.5; 2.51–259), and 3 (15.9; 1.30–195 (Table 6).

## 4. Discussion

This study aimed at identifying and comparing clinical characteristics of patients attending three hospitals for COVID-19 diagnosis and/or treatment at different stages of disease severity.

Overall, a gradient across COVID-19 groups was observed as the severity of the infection increased, with a higher prevalence of clinical events, including signs/symptoms and laboratory indicators, consistent with the reported results in recent meta-analyses [29,30].

Men accounted for two out of three patients hospitalized or intubated. Older age was also associated with being hospitalized or intubated, and with having comorbidities, especially DM2, HTN, and obesity, present in 33.3%, 50%, and 41.7% of the intubated patients, respectively. These findings were expected, as advanced age, male sex, and obesity have been reported as independent markers of poor prognosis [31,32]. The prevalence of general symptoms, such as fever, myalgia, arthralgia, headache, and fatigue, increased with COVID-19 severity, reaching 55–70% among the intubated. A similar gradient was seen for respiratory symptoms, including cough and dyspnea, prevalent in 83% of patients hospitalized or intubated. Our results follow a similar pattern of a meta-analysis of 3600 patients from 43 studies comparing critical vs. non-critical illness: namely, the higher prevalence of symptoms in critical patients (i.e., fever 80.8 vs. 71.2%, cough 65.6 vs. 56.7%, dyspnea 49.2 vs. 13.3%, fatigue 41.2 vs. 34.5%, myalgia 17.6 vs. 20.8%, and headache 11.3 vs. 11.9%) [33].

For the laboratory results, the increased trends with COVID-19 severity group were also seen. Close to 80% of the intubated patients had lymphopenia, hypertriglyceridemia, neutrophilia, and hyperglycemia. Lymphopenia has been consistently reported and associated with severity [19,33]; in fact, it has been said that lymphocyte count level can predict recovery or death [34]. Neutrophilia has been reported in ICU cases [35] or in patients with severe respiratory distress [36], and hyperglycemia has also been seen, but mostly in critically ill patients [37]. Hyperuricemia, hypercreatininemia, thrombocytopenia, and anemia followed a gradient with COVID-19 severity too, but to a lower extent. In this regard, low thrombocyte counts and high creatinine values have been documented, but mostly in severe cases [19,29,33].

In CHI, most intubated patients had elevated CRP (>50 mg/L), SF (>400 ng/mL), IL-6 (>7 pg/mL), and PCT (>0.5 ng/mL) values. Studies have reliably reported increased CRP values in all COVID-19 patients [19,33]; actually, there seems to be a correlation between CRP level and severity and prognosis, whereby survivors have lower median values (40 mg/L) than non-survivors (125 mg/L) [38]. Likewise, there is evidence associating higher SF levels among non-survivors [39]. Increased IL-6 values have been seen from mild to critical cases in several studies [19], and PCT has also been elevated in severe cases [29]. The fact that virtually all patients from groups 3 and 4 had CO-RADS 5 or 6 was expected, as this reporting system has shown quite good diagnostic performance in symptomatic individuals [40,41,42].

Most of the variables mentioned above also showed an increased gradient in crude multinomial logistic regression, with the highest ORs in the intubated group. The effect remained significant in multivariate analyses for male sex, DM2, obesity, neutrophilia, and for symptoms such as fever, myalgia/arthralgia, cough, and dyspnea; anosmia and dysgeusia showed an inverse effect, with higher odds among non-hospitalized patients. This final model explained 76% of the variability. Some of these variables were also found to be more frequent in patients with a severe infection in two recent meta-analyses [29,30]. In the one that included 3975 patients (1172 severe, 2803 non-severe) from 16 studies, those with a severe condition were more likely to have dyspnea and DM2 [29]; however, no statistical significance was reached in the adjusted model for lymphopenia, thrombocytopenia, or hypercreatininemia, which have been mainly documented among severe patients [29]. The other meta-analysis with 2445 patients (1,966 non-severe; 479 with severe illness or admitted to ICU) from 12 cohorts reported increased ORs for DM2 (OR 3.17; 95% CI 2.26–4.45), fever (1.67; 1.15–2.42), dyspnea (4.17; 2.04–8.53), but marginally for cough (1.26; 0.95–1.66) and myalgia (1.31; 0.97–1.77); lymphopenia, thrombocytopenia, hypercreatininemia, and elevated CRP were also more prevalent in the severe group [30]. The finding that patients from group 2 had higher odds for anosmia/dysgeusia compared with all other groups agrees with the results from a systematic review looking at olfactory deterioration and gustatory symptoms that showed a higher prevalence at earlier stages of the disease [43].

Finally, the main findings of this study match the symptomatology reported in a recent study that used a non-probabilistic sample of 1148 SARS-CoV-2 patients from one of the largest hospitals in Mexico. The symptoms with the highest associations for SARS-CoV-2 included anosmia, fever, dyspnea, and cough, all of which were statistically significant in the adjusted multinomial regression model presented here [44]. 

The study strengths include the collection of primary data from a fair sample of 342 individuals from the general population obtained during the second wave of the pandemic. The inclusion of a PCR-negative control group was also valuable to be able to have a reference comparison category for the crude and adjusted regression analyses. On the other hand, the participation of patients from three different hospitals located in central and northern Mexico improves the external validity of the results, making it possible to generalize the findings to populations and locations of similar characteristics (i.e., middle–high income settings from Latin America). Finally, the multinomial regression modeling was used, which fitted well with the grouping distribution as a dependent variable.

On the other hand, some limitations need to be kept in mind when interpreting the results presented. The COVID-19 grouping, used as a proxy for disease severity, could have led to misclassification bias; namely, patients that should have been hospitalized or intubated, but that were not due to insufficient hospital and ICU beds. The cross-sectional design used precluded a longitudinal follow-up of the clinical progression of the infection; we only relied on measures taken right after hospitalization. Another limitation relates to the use of group 1 as a reference, as it was unknown whether these individuals were sick (other than COVID-19) or healthy; in the latter case, estimates would result in significant differences. The fact that the hospitals included represent health care systems with different resources (i.e., private, social security, and public for uninsured people) that relates to differential diagnostic and therapeutic protocols could have also led to group misclassification. Finally, there was the incomplete assessment of some sociodemographic, clinical, and laboratory data in ZAC and SLP that could not be used in the regression analyses.

At present, the patients’ oxygen saturation level, number and severity of comorbidities, laboratory results, and pulmonary CT scan assessment are triage tools to determine hospital admission and treatment. At the same time, some of these indicators are used for infection staging, risk stratification, and prognosis. Therefore, the results presented here reinforce the diagnostic and prognostic value of several indicators of severity reported earlier. Future meta-analyses should consider running meta-regression to better identify significant clinical, laboratory, and radiologic features of COVID-19.

## Figures and Tables

**Figure 1 healthcare-09-00895-f001:**
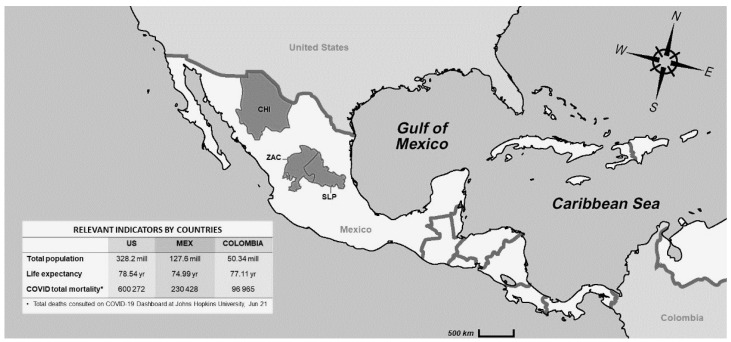
Study locations, and comparison of basic indicators (population, life expectancy, and COVID mortality) with the United States and Colombia.

**Table 1 healthcare-09-00895-t001:** Characteristics of the participant hospitals, patients, samples collected and type of analysis for the multicentric COVID-19, Mexico 2020.

City	Hospital Characteristics	Group Sample ^a^	Collection Dates	Inclusion Criteria	Data Obtained
Chihuahua, 1 million inhabitants	Christus Murguerza Private, general 60 beds (19 COVID) 10 ICU (5 COVID)	1) 49 2) 29 3) 37 4) 12 Total = 127	15 August 2020 to 1 December 2020	Age > 18 y Informed consent	Clinical, *n* = 127 Lab, *n* = 52 CT, *n* = 49
San Luis Potosí 300,000 inhabitants	Soledad Graciano Public, general90 beds (90 COVID) 8 ICU (8 COVID)	1) 5 2) 3 3) 39 4) 8 Total = 55	15 September 2020 to 1 December 2020	Age > 18 y Informed consent	Clinical, *n* = 55 Lab, *n* = 43
Zacatecas, 200,000 inhabitants	General Hospital Public, institutional 207 beds (95 COVID) 10 ICU (5 COVID)	1) 40 2) 40 3) 40 4) 40 Total = 160	15 March 2020 to 1 December 2020	Age 35–70 y	Clinical, *n* = 160 Lab, *n* = 139

^a^ Groups: 1) PCR −, controls; 2) PCR +, not hospitalized; 3) PCR +, hospitalized with/without supplementary oxygen; and 4) PCR +, intubated at the intensive care unit.

**Table 2 healthcare-09-00895-t002:** Socio-demographic characteristics, main comorbidities, symptomatology and epidemiological data of participants stratified by hospital location, multicentric COVID-19 study, Mexico 2020.

Variable	Category	Frequency (%)			
		CHI *n* = 127	SLP *n* = 55	ZAC *n* = 160	Total *n* = 342
Sex	Male	80 (63.0)	39 (70.9)	88 (55.0)	207 (60.5)
	Female	47 (37.0)	16 (29.1)	72 (45.0)	135 (39.5)
Age in years	Mean ± s.d	43.3 ± 14.5	53.6 ± 15.09	53.2 ± 10.2	49.6 ± 13.6
Age group in years	20–40	62 (48.8)	11 (20.0)	29 (18.1)	102 (29.8)
	41–50	26 (20.5)	14 (25.5)	34 (21.3)	74 (21.6)
	51–60	26 (20.5)	14 (25.5)	53 (33.1)	93 (27.2)
	61–70	4 (3.1)	9 (16.4)	44 (27.5)	57 (16.7)
	>70	9 (7.1)	7 (12.7)	0 (0.0)	16 (4.7)
Civil status	Single	36 (28.8)	-	-	36 (28.8)
	Married/free union	76 (60.8)	-	-	76 (60.8)
	Divorced/separated	9 (7.2)	-	-	9 (7.2)
	Widow(er)	4(3.2)	-	-	4(3.2)
Occupation	Home	18 (14.3)	-	-	18 (14.3)
	Employed	98 (77.8)	-	-	98 (77.8)
	Student	3 (2.4)	-	-	3 (2.4)
	Retired	7 (5.6)	-	-	7 (5.6)
Physical activity	Sedentary life	72 (58.5)	-	-	72 (58.5)
	2–3 days per week	21 (17.1)	-	-	21 (17.1)
	Every day	30 (24.4)	-	-	30 (24.4)
Current smoking		22 (17.3)	2 (3.6)	15 (9.4)	39 (11.4)
Type 2 diabetes		12 (9.4)	23 (41.8)	38 (23.8)	73 (21.4)
Hypertension		28 (22.0)	20 (36.4)	53 (33.1)	101 (29.5)
COPD or asthma		9 (7.1)	3 (5.5)	6 (3.8)	18 (5.3)
Immunosuppressed		5 (3.9)	0 (0.0)	2 (1.3)	7 (2.1)
Chronic kidney dis.		4 (3.1)	2 (3.6)	5 (3.2)	11 (3.2)
Obesity	BMI ≥ 30 kg/m^2^	43 (33.9)	21 (38.2)	39 (24.4)	103 (30.1)
BMI (kg/m^2^)	Mean ± s.d	29 ± 6.8	-	-	29 ± 6.8
	18.5–24.9	23 (21.5)	-	-	23 (21.5)
	25–30	41 (38.3)	-	-	41 (38.3)
	>30	43 (40.2)	-	-	43 (40.2)
General symptoms	Fever	35 (27.8)	38 (30.9)	69 (43.6)	142 (41.6)
	Headache	31 (24.6)	36 (65.5)	107 (66.9)	174 (51)
	Myalgia	48 (38.1)	32 (58.2)	83 (51.9)	163 (47.8)
	Arthralgia	45 (35.7)	30 (54.5)	76 (47.5)	151 (44.3)
	Fatigue	53 (42.1)	35 (63.6)	62 (38.8)	150 (44)
Respirat. symptoms	Cough	48 (38.1)	41 (74.5)	97 (60.6)	186 (54.5)
	Sore throat	16 (12.7)	25 (45.5)	63 (39.4)	104 (30.5)
	Dyspnea	44 (34.9)	46 (83.6)	83 (51.9)	173 (50.7)
	Chest pain	13 (10.3)	31 (56.4)	35 (21.9)	79 (23.2)
Other signs/symp.	Anosmia/dysgeusia	9 (7.1)	23 (14.4)	32 (11.2)	32 (11.2)
	Diarrhea	13 (10.3)	15 (27.3)	19 (11.9)	47 (13.8)
Immunosup. drugs ^a^	Dexamethasone	44 (34.9)	-	-	44 (34.9)
	Baricitinib	24 (19.0)	-	-	24 (19.0)
	Tocilizumab	7 (5.6)	-	-	7 (5.6)
	HCQ/azithromycin	5 (4.0)	-	-	5 (4.0)
Epidemiol. data	COVID contact	72 (57.1)	-	65 (40.6)	137 (47.9)
	Influenza vaccine	28 (22.2)	13 (23.6)	24 (15.4)	65 (19.3)
	Recent travel	15 (11.9)	-	48 (16.8)	48 (16.8)

^a^ Administered prior to the blood/urine sample collection.

**Table 3 healthcare-09-00895-t003:** Socio-demographic characteristics, main comorbidities and symptomatology for participants stratified by group, multicentric COVID-19 study, Mexico 2020.

Indicator ^b^	Measure, Unit	Group ^a^, Frequency (%)			
		1 (*n* = 94)	2 (*n* = 72)	3 (*n* = 116)	4 (*n* = 60)
Sex	Male	46 (48.9)	41 (56.9)	79 (68.1)	41 (68.3)
	Female	48 (51.1)	31 (42.1)	37 (31.9)	19 (31.7)
Age in years	Mean ± s.d	41.1 ± 11.7	47.5 ± 13.5	54.8 ± 12.3	55.3 ± 11.5
Age group in years	20–40	56 (16.4)	26 (36.1)	12 (10.3)	8 (13.3)
	41–50	17 (18.1)	10 (13.9)	38 (15.0)	9 (15.9)
	51–60	14 (4.1)	22 (30.6)	34 (29.3)	23 (27.2)
	61–70	6 (6.4)	14 (19.4)	19 (16.4)	18 (30.0)
	>70	1 (1.1)	0 (0.0)	13 (11.2)	2 (3.3)
Current smoking		10 (10.6)	11 (3.2)	13 (11.2)	5 (1.5)
Type 2 diabetes		5 (5.3)	5 (6.9)	43 (37.1)	20 (33.3)
Hypertension		16 (17.0)	14 (19.4)	41 (35.3)	30 (50.0)
COPD or asthma		4 (4.3)	3 (4.2)	7 (6.0)	4 (6.7)
Immunosuppressed		3 (3.2)	1 (1.4)	3 (2.6)	0 (0.0)
Chronic kidney dis.		1 (1.1)	1 (1.4)	6 (5.2)	3 (5.0)
Obesity (BMI ≥ 30)		20 (21.3)	13 (18.1)	45 (38.8)	25 (41.7)
General symptoms	Fever	1 (1.1)	27 (37.5)	73 (62.9)	41 (68.3)
	Headache	37 (39.8)	37 (51.4)	66 (56.9)	34 (56.9)
	Myalgia	23 (24.7)	29 (40.3)	69 (59.5)	42 (70.0)
	Arthralgia	14 (15.1)	29 (40.3)	68 (58.6)	40 (66.7)
	Fatigue	15 (16.1)	23 (31.9)	79 (68.1)	33 (55.0)
Respiratory symptoms	Cough	11 (11.8)	40 (55.6)	85 (73.3)	50 (83.3)
	Sore throat	21 (22.6)	22 (30.6)	35 (30.2)	26 (43.3)
	Dyspnea	8 (8.6)	13 (18.1)	102 (88)	50 (83.3)
	Chest pain	8 (8.6)	11 (15.3)	42 (36.2)	18 (30.0)
Other signs/symptoms	Anosmia/dysgeusia	1 (1.1)	12 (17.4)	14 (18.2)	5 (9.6)
	Diarrhea	5 (5.4)	8 (11.1)	24 (20.7)	10 (16.7)

^a^ Groups: (1) PCR −, controls; (2) PCR +, not hospitalized; (3) PCR +, hospitalized with/without supplementary oxygen; and (4) PCR +, intubated at the intensive care unit; ^b^ Only data from variables assessed in all 3 hospitals are presented (total sample for all symptoms in group 1 was 93; for anosmia/dysgeusia, samples were 88, 69, 77, and 52 for groups 1, 2, 3, and 4, respectively).

**Table 4 healthcare-09-00895-t004:** Laboratory and radiologic findings for participants stratified by COVID-19 group, multicentric COVID-19 study, Mexico 2020.

Indicator	Measure or Unit	Cut-Off	Group ^a^, Frequency/Sample (%)			
			1	2	3	4
↓ hemoglobin	g/dL	Anemia ^b^	3/45 (6.7)	5/34 (14.7)	29/98 (29.6)	16/55 (29.1)
↑ neutrophils	×10^3^/L	>7.5	5/45 (11.1)	8/34 (23.5)	67/98 (68.6)	43/54 (79.6)
↓ lymphocytes	×10^3^/L	<1.5	7/45 (15.6)	18/34 (52.9)	77/98 (78.6)	47/54 (87.0)
↑ lymphocytes	×10^3^/L	>3.5	2/45 (4.4)	2/34 (5.9)	0/98 (0)	0/54 (0)
↓ platelets	×10^3^/L	<150	1/45 (2.2)	3/34 (8.8)	8/98 (8.2)	8/55 (14.5)
↓ glucose	mg/dL	<60	0/46 (0)	0/33 (0)	2/99 (2)	2/55 (3.6)
↑ glucose	mg/dL	>100	16/46 (34.8)	17/33 (51.5)	75/99 (75.8)	43/55 (78.2)
↑ creatinine	mg/dL	>1.2	5/45 (11.1)	1/34 (2.9)	14/99 (14.1)	20/56 (35.7)
↑ uric acid	mg/dL	>50	2/27 (7.4)	1/15 (6.7)	23/90 (25.6)	27/45 (60.0)
↑ cholesterol	mg/dL	>200	1/8 (12.5)	-	1/43 (2.3)	1/19 (5.3)
↑ triglycerides	mg/dL	>150	2/7 (28.6)	-	21/43 (48.8)	16/19 (84.2)
For CHI only						
↑ PCT	ng/mL	>0.5	0/2 (0)	-	5/35 (14.3)	3/11 (27.3)
↑ CRP	mg/L	>50	2/2 (100)	-	33/33 (100)	7/10 (70.0)
↑ SF	ng/mL	>400	0/1 (0)	-	27/34 (79.4)	6/6 (100)
↑ IL-6	pg/mL	>7	-	-	11/12 (91.7)	5/5 (100)
IgG, COVID	Positive		5/48 (10.4)	0/29 (0)	10/37 (27.0)	1/12 (8.3)
IgM, COVID	Positive		8/53 (15.1)	3/32 (9.4)	19/76 (25.0)	2/20 (10.0)
CT scan, *n* (%)						
CORADS	1	-	-	-	-	-
	2	-	-	-	-	-
	3	-	-	-	-	-
	4	-	-	-	3/37 (8.1)	0/12 (0)
	5	-	-	-	29/37 (78.4)	7/12 (58.3)
	6	-	-	-	5/37 (13.5)	5/12 (41.7)

^a^ Groups: (1) PCR − controls, (2) PCR +, not hospitalized, (3) PCR +, hospitalized with/without supplementary oxygen, and (4) PCR +, intubated at the intensive care unit; ^b^ Females <12 g/dL, males <14 g/dL. CHI: Chihuahua, PCT: procalcitonin, CRP: C-reactive protein, SF: serum ferritin, IL-6: interleukin 6, IgG: Immunoglobulin G, IgM: Immunoglobulin M, CORADS: COVID-19 reporting and data system.

**Table 5 healthcare-09-00895-t005:** Crude odds ratio (OR) with 95% confidence intervals (CIs) from multinomial logistic regression with severity groups as dependent variable (group 1 as reference) and selected independent variables, multicentric COVID-19 study, Mexico 2020.

Variable	Category	Groups ^a^, Crude OR (95% CI)		
		2 vs. 1	3 vs. 1	4 vs. 1
Sex	Female	1.00	1.00	1.00
	Male	1.38 (0.74–2.55)	2.22 (1.27–3.90)	2.25 (1.14–4.43)
Age group in years	20-45	1.00	1.00	1.00
	46-65	2.76 (1.43–5.35)	6.99 (3.66–13.3)	5.41 (2.54–11.6)
	>65	2.00 (0.46–8.51)	12.8 (3.97–41.1)	12.8 (3.61–45.2)
Pulmonary disease	yes vs. no	0.97 (0.21–4.51)	1.44 (0.41–5.09)	1.60 (0.38–6.68)
Current smoking	yes vs. no	1.51 (0.60–3.79)	1.06 (0.44–2.53)	0.76 (0.24–2.35)
Type 2 diabetes	yes vs. no	1.32 (0.37–4.77)	10.4 (3.94–27.8)	8.90 (3.10–25.3)
Hypertension	yes vs. no	1.17 (0.53–2.60)	2.66 (1.37–5.15)	4.87 (2.32–10.2)
Immunosuppressed	yes vs. no	0.42 (0.04–4.19)	0.81 (0.16–4.12)	ND
Chronic kidney disease	yes vs. no	1.31 (0.08–21.3)	5.07 (0.60–42.8)	4.89 (0.49–48.1)
Obesity	yes vs. no	0.81 (0.37–1.77)	2.34 (1.26–4.35)	2.64 (1.29–5.38)
Physical activity	Daily	1.00	1.00	1.00
	2–3 times/wk	1.09 (0.28–4.19)	4.66 (0.91–23.7)	1.00 (0.07–13.0)
	Sedentary	0.63 (0.22–1.82)	5.25 (1.34–20.5)	2.62 (0.49–13.9)
Anemia	yes vs. no	2.41 (0.53–10.9)	5.88 (1.68–20.5)	5.74 (1.55–21.5)
↑ neutrophils	yes vs. no	2.46 (0.72–8.35)	17.2 (6.21–48.0)	31.2 (9.98–97.2)
↓ lymphocytes	yes vs. no	6.61 (2.27–19.2)	18.8 (7.34–48.3)	6.61 (2.27–19.2)
↑ glucose	yes vs. no	1.99 (0.79–4.96)	6.39 (2.95–13.8)	8.06 (3.22–20.1)
↑ creatinine	yes vs. no	0.24 (0.02–2.17)	1.31 (0.44–3.91)	4.44 (1.51–13.0)
↑ triglycerides	yes vs. no	ND	0.16 (0.00–2.98)	0.38 (0.02–7.11)
Fever	yes vs. no	55.2 (7.26–419)	156 (21.0–1161)	198 (25.7–1533)
Headache	yes vs. no	9.31 (4.26–20.3)	20.4 (9.63–43.3)	37.2 (14.7–94.0)
Myalgia	yes vs. no	2.05 (1.05–3.99)	4.46 (2.45–8.13)	7.10 (3.43–14.6)
Arthralgia	yes vs. no	3.80 (1.81–7.96)	7.99 (4.05–15.7)	11.2 (5.16–24.6)
Fatigue	yes vs. no	2.40 (1.10–5.10)	11.1 (5.60–21.8)	6.30 (2.90–13.4)
Cough	yes vs. no	9.63 (4.20–20.3)	20.4 (9.60–43.3)	37.2 (14.7–94.0)
Sore throat	yes vs. no	1.50 (0.75–3.03)	1.48 (0.79–2.77)	2.62 (1.29–5.30)
Dyspnea	yes vs. no	2.34 (0.91–−6.00)	77.4 (31.0–193)	53.1 (19.6–143)
Chest pain	yes vs. no	1.91 (0.72–5.04)	6.03 (2.66–13.6)	4.55 (1.83–11.3)
Anosmia/dysgeusia	yes vs. no	18.3 (2.30–144)	19.3 (2.40–150)	9.25 (1.00–150)
Diarrhea	yes vs. no	2.20 (0.68–7.03)	4.59 (1.67–12.5)	3.52 (1.13–10.8)

^a^ Groups: (1) PCR −, controls; (2) PCR +, not hospitalized; (3) PCR +, hospitalized with/without supplementary oxygen; and (4) PCR +, intubated at the intensive care unit. ND: No data for this group/category.

**Table 6 healthcare-09-00895-t006:** Adjusted odds ratio (OR) with 95% confidence intervals (CIs) and *p*-values from multinomial logistic regression with groups as dependent variable (group 1 as reference) and selected independent variables, multicentric COVID-19 study, Mexico 2020.

Variable	Category	Group ^a^, Adjusted OR (95% CI) ^b^		
		2 vs. 1	3 vs. 1	4 vs. 1
Sex	Female	1.00	1.00	1.00
	Male	1.90 (0.87–4.15) *p* = 0.10	3.34 (1.13–9.89) *p* = 0.02	3.66 (1.12–11.9) *p* = 0.03
Type 2 diabetes	yes vs. no	1.56 (0.29–8.16) *p* = 0.38	12.8 (2.50–65.8) *p* = 0.002	16.1 (2.87–90.2) *p* = 0.002
Obesity	yes vs. no	0.79 (0.31–2.05) *p* = 0.64	3.38 (1.04–10.9) *p* = 0.04	4.10 (1.16–14.4) *p* = 0.02
Fever	yes vs. no	45.5 (4.55–454) *p* = 0.001	49.2 (4.61–525) *p* = 0.001	62.9 (5.60–707) *p* = 0.001
Myalgia/arthralgia	yes vs. no	0.64 (0.24–1.73) *p* = 0.38	2.82 (0.82–9.68) *p* = 0.09	4.31 (1.14–16.2) *p* = 0.03
Cough	yes vs. no	4.43 (1.72–11.3) *p* = 0.002	10.5 (3.00–36.8) *p* < 0.000	26.4 (6.40–109) *p* < 0.000
Dyspnea	yes vs. no	1.04 (0.32–3.37) *p* = 0.94	27.07 (7.31–100) *p* < 0.000	21.3 (5.03–90.5) *p* < 0.000
Anosmia/dysgeusia	yes vs. no	25.5 (2.51–259) *p* = 0.02	15.9 (1.30–195) *p* = 0.03	6.87 (0.48–96.8) *p* =0.15
↑ neutrophils	yes vs. no	1.51 (0.33–6.83) *p* = 0.59	6.71 (1.46–30.6) *p* = 0.01	16.5 (3.26–84.1) *p* = 0.001

^a^ Groups: (1) PCR −, controls; (2) PCR +, not hospitalized; (3) PCR +, hospitalized with/without supplementary oxygen; and (4) PCR +, intubated at the intensive care unit; ^b^ Variables with *p* < 0.10 in crude analyses were entered in the full model, but only those with at least one statistically significant (*p* < 0.05) category in the comparisons remained in the final model (age group, hospital location, hypertension, fatigue, sore throat, chest pain, diarrhea, headache, anemia, lymphopenia, hypercreatininemia, and hyperglycemia, were excluded); Nagelkerke = 0.764.

## Data Availability

Data is available from the first author upon request.

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
