# Peer review of "Clinical Factors Associated with COVID-19 Severity in Mexican Patients: Cross-Sectional Analysis from a Multicentric Hospital Study"

_healthcare, 2021, doi:10.3390/healthcare9070895_

Round 1

Reviewer 1 Report

This study describe the association of several variables with COVID 19 in patients cathegorized in 4 groups based on the severity of the disease. The article is of great scientific and social impact and it provides new insight in this new disease.

The authors should divide “Results” paragraph in at least 2 paragraphs: a first paragraph on the descriptive analysis of  groups based on hospital and based on disease severity (from line 175 to 217), a second paragraph  on the association analysis (fromline 218 to240). In alternative the authors could divide the results paragraph in 4 paragraphs: the first on the general characteristic of population of each hospital (from line 175 to 195), the second on the general characteristic of population in each disease groups (from line 198 to 206), the third on the laboratory and radiologic analysis, and the  fourth on the association analysis (from line 219 to 239).

Author Response

Mortality data presented in the Introduction section was updated. Changes requested by reviewers 2 and 3 were also made.

We were unsure about the reviewer's request, as the results section was already divided according to the tables' contents. However, attending the reviewer's concern, we decided to include five subheadlines to highlight the contents of each table. We also introduced each subsection with a sentence presenting the contents of the tables. Finally, we divided long paragraphs into shorter ones based on the different topics covered in the table (e.g. sociodemographic data, symptomatology, etc.)

Reviewer 2 Report

It’s a comprehensive study across three different hospitals

 Recommendations: the discussion should end with conclusions from the study.

Grammar could be improved

Author Response

Mortality data presented in the Introduction section was updated. English language, grammar, and style were checked and corrected throughout the manuscript. No changes were possible to improve the design, as the study is now completed. In the methods section we included Figure 1 depicting the study location and comparing basic indicators with two other countries from the region, as specifically requested by reviewer 3. The results were reorganized in five sections using sub headlines, as requested by reviewer 1; a sentence introducing the contents of each table was added, and long paragraphs were split to improve the reading of the results based on the topics presented. Finally, the discussion section now ends with the paragraph of conclusions.

Reviewer 3 Report

Reviewer report

Dear Author(s):

In accordance with the review of the article "Clinical factors associated with COVID-19 severity in Mexican patients: Cross-sectional analysis from a multicentric hospital study" (healthcare-1266477), which has been assigned to my person, I communicate below what my decision has been:

Accept after minor revision (corrections to minor methodological errors and text editing).

First of all, this reviewer has to congratulate the work done by the authors, demonstrating a wide knowledge of the subject analyzed. However, I would consider it necessary that the authors perform the following "tweaks" in order that their paper gains some more added value.

  1. They should indicate the geographic distance of each of the hospital units studied.
  2. They should include a map of Mexico in which each one of them is indicated.
  3. Regardless of the analysis performed, it would not be superfluous to include a table with a series of key socioeconomic magnitudes of Mexico that are intimately related to the work conducted, such as life expectancy at birth, life expectancy at birth, life expectancy at birth, life expectancy at birth, life expectancy at birth, life expectancy at birth, level of education, unemployment rate, type of work performed by productive sector, etc, and, of course, relate it to the results obtained.
  4. Finally, in the final discussions, I would ask you to relate the results obtained with the pre-existing literature and to suggest the possibility of carrying out this same study in countries geographically and socioculturally close to Mexico, especially in the Mesoamerican area.

With my best wishes in your academic and personal life,

The reviewer

Author Response

Mortality data presented in the Introduction section was updated. English language, grammar, and style were checked and corrected throughout the manuscript. No changes were possible to improve the design, as the study is now completed. In the Methods section we added a map (Figure 1) and compared basic indicators with two other countries from the region (United States and Colombia), as requested by the reviewer. As to the Discussion section, we have already cited more than a dozen articles to contextualize our findings, including four systematic reviews/meta-analyses. However, we added a new article from Mexico (published in March 2021) that presents relevant clinical symptomatolgy, in line with the findings of our study. Finally, in this revised version we now discuss more in detail the strenghts of the study, including the external validity of the results at request of the reviewer.
